# Inverse Association between the Existence of CRISPR/Cas Systems with Antibiotic Resistance, Extended Spectrum β-Lactamase and Carbapenemase Production in Multidrug, Extensive Drug and Pandrug-Resistant *Klebsiella pneumoniae*

**DOI:** 10.3390/antibiotics12060980

**Published:** 2023-05-29

**Authors:** Noor A. Jwair, Mushtak T. S. Al-Ouqaili, Farah Al-Marzooq

**Affiliations:** 1Department of Microbiology, College of Medicine, University of Anbar, Ramadi P.O. Box 55431, Iraq; noo20m0019@uoanbar.edu.iq; 2Department of Microbiology and Immunology, College of Medicine and Health Sciences, United Arab Emirates University, Al-Ain P.O. Box 15551, United Arab Emirates; f.almarzooq@uaeu.ac.ae

**Keywords:** CRISPR/Cas, *K. pneumoniae*, extended-spectrum β-lactamases, carbapenemases, antibiotic resistance

## Abstract

Antimicrobial resistance, with the production of extended-spectrum β-lactamases (ESBL) and carbapenemases, is common in the opportunistic pathogen, *Klebsiella pneumoniae*. This organism has a genome that can contain clustered regularly interspaced short palindromic repeats (CRISPRs), which operate as a defense mechanism against external invaders such as plasmids and viruses. This study aims to determine the association of the CRISPR/Cas systems with antibiotic resistance in *K. pneumoniae* isolates from Iraqi patients. A total of 100 *K. pneumoniae* isolates were collected and characterized according to their susceptibility to different antimicrobial agents. The CRISPR/Cas systems were detected via PCR. The phenotypic detection of ESBLs and carbapenemases was performed. The production of ESBL was detected in 71% of the isolates. Carbapenem-resistance was detected in 15% of the isolates, while only 14% were susceptible to all antimicrobial agents. Furthermore, the bacteria were classified into multidrug (77%), extensively drug-resistant (11.0%) and pandrug-resistant (4.0%). There was an inverse association between the presence of the CRISPR/Cas systems and antibiotic resistance, as resistance was higher in the absence of the CRISPR/Cas system. Multidrug resistance in ESBL-producing and carbapenem-resistant *K. pneumoniae* occurred more frequently in strains negative for the CRISPR/Cas system. Thus, we conclude that genes for exogenous antibiotic resistance can be acquired in the absence of the CRISPR/Cas modules that can protect the bacteria against acquiring foreign DNA.

## 1. Introduction

In recent times, the most urgent global problem for public health is the rapid emergence of antibiotic-resistant superbugs [1]. One of these superbugs is the extensively drug-resistant (XDR) and pan-drug-resistant (PDR) *Klebsiella pneumoniae*, a Gram-negative bacterium with bacillary morphology, belonging to the order *Enterobacterales* [2,3]. The vast majority of antibiotic-resistant genes found in *K. pneumoniae* are shared with other Enterobacteriaceae, primarily via self-transferrable plasmids that are considered the major contributors to antibiotic resistance in many bacteria [4,5].

Two of the main resistance mechanisms developed by *K. pneumoniae* are the production of extended spectrum β-lactamase (ESBL) and carbapenemase enzymes [6]. Different variants of these enzymes are capable of inhibiting the action of various β-lactams, including cephalosporins, penicillins, monobactams and carbapenems, which are among the most prescribed antibiotics globally [6,7]. Infections caused by ESBL-producing *K. pneumoniae* are typically treated with carbapenems. However, due to the emergence of carbapenem-resistant *K. pneumoniae* in recent years, other drugs such as tigecycline and colistin have replaced carbapenems as the only available therapeutic option to treat infections caused by XDR bacteria [2,8].

In several bacteria, clustered regularly spaced short palindromic repeats (CRISPRs) and their associated genes (Cas) act as a defense mechanism against foreign genomes such as plasmids and resistance genes [9,10]. A CRISPR/Cas system is made up of upstream Cas genes, a variety of exogenous DNA sequences (known as spacers), and similarly sized and identical direct and inverted repeats (known as palindromic repeats) [9,10,11]. The origin of these spacers is the invasion of foreign plasmids and/or phages into the bacterial cell, and their chronological insertion into the CRISPR array denotes the acquisition of a “memory fragment” of such invaders. The bacteria precisely identify the DNA sequences matching its spacers and cleave them through the nuclease activity of the Cas proteins, acting as a sequence-specific bacterial defense [9,10,11].

In the last decade, an inverse correlation between the presence of the CRISPR/Cas system in the bacterial genome, including *K. pneumoniae,* and antibiotic resistance has been demonstrated in numerous studies [4,9]. However, the results in this field have sometimes been contradictory, so the need for more research in this area is quite apparent [12,13]. Understanding the association between the existence of the CRISPR loci and antibiotic resistance determinants, including ESBLs and carbapenemases in *K. pneumoniae,* may provide new insights into new drug targets to fight infections caused by this drug-resistant organism. Hence, this study aimed to determine the association of the CRISPR/Cas systems with antibiotic resistance, extended spectrum β-lactamase and carbapenemase production in *K. pneumoniae* isolates from Iraqi patients.

## 2. Results

### 2.1. Susceptibility to Antimicrobial Agents

One hundred isolates of *K. pneumoniae* were isolated from a variety of clinical specimens. All the bacterial isolates were tested for susceptibility to 15 different antibiotics (Figure 1). The isolates showed a high resistance to several antibiotics with different mechanisms of action (cell wall synthesis inhibitors, DNA synthesis inhibitors and protein synthesis inhibitors).

As shown in Figure 1, the majority of the isolates were resistant to multiple beta-lactam drugs including carbapenems and beta-lactam/beta lactamase inhibitors. Furthermore, resistance was common to other drugs such as fluoroquinolones and aminoglycosides. The majority of the strains were susceptible to tigecycline, with only a few resistant strains (6%).

Based on their susceptibility to different antibiotics and the results of different phenotypic tests, the production of ESBL was detected in 71% (71/100) of isolates. All ESBL-producing strains were multidrug-resistant (MDR). Furthermore, carbapenem-resistance (CR) was detected in 15% (15/100) of isolates, and all of them were metallo-β-lactamase (MBL) producers. From CR, four were PDR and the rest (*n* = 11) were XDR. Only 14% (14/100) of the strains were susceptible to almost all antimicrobial agents without MDR phenotype.

### 2.2. Distribution of CRISPR/Cas System Elements among K. pneumoniae Isolates

PCR was used for the detection of Cas1, Cas3, CRISPR1, CRISPR2 and CRISPR3 genes. Table 1 shows the different profiles detected in the 100 strains based on the presence or absence of Cas and/or CRISPR genes. Representative gel images showing CRISPR and Cas elements in *K. pneumoniae* isolates are shown in Appendix A.

As shown in Table 1, some strains (profile 2–7) carried one component of the system (either CRISPR or Cas), while others had both components (profile 8–23); thus, the CRISPR/Cas system (s) was detected in 32% of the isolates. Different combinations of the genes were detected as shown in the table.

It is noteworthy that the majority of the 49 strains having either the CRISPR or Cas alone were ESBL-MDR (*n* = 34) or CR strains (*n* = 9), compared to only 6 susceptible strains without MDR phenotype (*p* < 0.05).

A total of 19 strains (19%) were negative for both CRISPR and Cas. These strains were either ESBL-MDR (*n* = 17) or CR strains (*n* = 2), with an absence of any susceptible strains. As for the 32 strains positive for both CRISPR and Cas, they encompassed a mixture of MDR-ESBL (*n* = 20), CR (*n* = 4), and susceptible strains without the MDR phenotype (*n* = 8).

### 2.3. The Frequency of the CRISPR/Cas System in Nosocomial Isolates of K. pneumoniae and its Association with Antimicrobial Resistance

The CRISPR/Cas system was demonstrated to interfere with the transformation and stability of the plasmids that often carry drug-resistant genes. Accordingly, we assessed whether the presence of CRISPR/Cas in clinical strains was related to their drug resistance.

The PCR revealed that 25 (67.57%) of the Cas1 PCR-positive strains also had at least one CRISPR array (CRISPR1, CRISPR2 or CRISPR3). The prevalence of the CRISPR/Cas system was four (26.67%) in carbapenem-resistant strains, which had a higher resistance to other antibiotics as these strains were either XDR or PDR. The prevalence of the CRISPR/Cas system was 21 (29.58%) in the ESBL-producing strains, while it was 8 (57.14%) in non-MDR strains, showing a significant (*p*-value > 0.05) inverse correlation between prevalence and resistance.

The low frequency of the CRISPR/Cas system in drug-resistant *K. pneumoniae* implied that the CRISPR/Cas may play a role in preventing the acquisition of drug-resistance genes. The occurrence of this system for the sensitive isolates was 6.0 (42.86%), 1.0 (7.14%) and 1.0 (7.14%) for CRISPR1/Cas, CRISPR2/Cas and CRISPR3/Cas, respectively. Whilst, by comparison, for the resistant isolates the same figures were 6.0 (8.45%), 19.0 (26.76%) and 13.0 (18.31%) for CRISPR1/Cas, CRISPR2/Cas and CRISPR3/Cas in ESBL isolates, respectively. As for the MBL-producing isolates; 1.0 (6.67%), 4.0 (26.67%) and 3.0 (20.0%) for CRISPR1/Cas, CRISPR2/Cas and CRISPR3/Cas were detected, respectively.

### 2.4. Association between Drug Resistance and Prevalence of the CRISPR/Cas System

The proportion of *K. pneumoniae* isolates that were resistant to antimicrobial agents were clearly higher in the CRISPR/Cas-negative isolates than in the CRISPR/Cas-positive isolates. The CRISPR/Cas-positive isolates showed resistance to piperacillin /tazobactam (18.75%), cefazolin (53.13%), cefoxitin (18.75%), ceftazidime (56.25%), ceftriaxone (53.13%), cefepime (56.25%), imipenem, ertapenem (12.5%), amikacin (12.5%), gentamicin (18.75%), ciprofloxacin / levofloxacin (25.0%), nitrofurantoin (12.5%) and trimethoprim/sulfamethoxazole (56.25%). In comparison, the resistance rates to those drugs in the CRISPR/Cas-negative isolates were 30.88%, 85.29%, -38.24%, 94.11%, 85.29%, 82.35%, 16.18%, 16.18%, 16.18%, 27.94%, 47.05%, 42.64%, 22.05% and 82.35%, respectively, in the isolates without the CRISPR/Cas system, as shown in Figure 2. It is obvious that the absence of CRISPR is more common among resistant strains, even if non-significant.

Our results showed that CRISPR2 is more common in resistant bacteria (*n* = 32; 37.2%), followed by CRISPR1 (*n* = 17; 19.77%) then CRISPR3 (*n* = 16; 18.6%), while in sensitive strains without MDR phenotype, CRISPR1 was most common (*n* = 10; 71.43%), followed by CRISPR3 (*n* = 3; 21.43%) then CRISPR2 (*n* = 2; 14.29%).

From Table 2, we can see that the most significant differences were obvious in the MDR and XDR groups as most of these strains were negative for CRISPR/Cas systems. As for non-MDR and PDR, the difference was not obvious, owing to the low total number of strains per group; thus, this limited the possibility of valid statistical comparisons. However, when we compare CRs as a full group (combining XDR and PDR), then it is very clear that the majority of these strains (*n* = 11) were negative for the CRISPR/Cas systems. The distribution of the CRISPR and Cas elements and systems in *K. pneumoniae* isolates with statistical comparisons are shown in Appendix A. A list of *K. pneumoniae* isolates, antibiotic susceptibility data and the CRISPR and Cas elements and systems detected in each strain are shown in Appendix A.

## 3. Discussion

Hospitals and healthcare facilities have recently emerged as the primary receptacles for a variety of multidrug-resistant (MDR) pathogenic bacteria, including species of *K. pneumoniae* [14]. The uncontrolled and unchecked use of antibiotics to treat infections is one of the causes of this phenomenon since it puts pressure on bacteria to evolve resistance. Indeed, the spread of antibiotic-resistant bacteria from hospitals and healthcare centers through various routes is playing an important role in spreading resistance to the community [15].

Bacteria which produce ESBLs are resistant to many penicillins and third-generation cephalosporin antibiotics. *Escherichia coli* and *Klebsiella* species are the main bacteria that produce ESBLs. A recent report by the Centers for Disease Control and Prevention in the USA indicates that nearly 26,000 healthcare-associated infections are caused by ESBL-producing Enterobacteriaceae every year [16]. Patients with bloodstream infections caused by ESBL-producing Enterobacteriaceae are about 57% more likely to die than those with bloodstream infections caused by non ESBL-producing strains [17].

This study investigated the resistance patterns of *K. pneumoniae* against different antibiotics, in which we assessed 100 clinical isolates of *K. pneumoniae* with different resistance status to determine the relationship between the CRISPR systems and antibiotic resistance. The study found that the prevalence of antibiotic resistance in *K. pneumoniae* is particularly high with regard to many important antimicrobial agents [18]. The study results revealed a high resistance rate to β-lactam antibiotics such as cefazolin (85.0%), ceftazidime (84.0%) and ceftriaxone (83.0%). This may be the result of the extensive use of these types of antibiotics by individuals in the absence of appropriate medical supervision. Mahmood and colleagues [19] revealed that the proportion of *K. pneumoniae* isolates which were resistant to the third-generation of cephalosporins was 88.63%.

The high level of resistant isolates corresponds to many local and international studies. These results were similar to those observed by Khalaf and Al-Ouqaili [8], who concluded that the rates of resistance to ceftriaxone and ceftazidime were 88% and 84%, respectively. Al-Kubaisy and colleagues [20] recorded that *K. pneumoniae* isolates were resistant to ceftriaxone and ceftazidime at rates of 88.2% and 82.3%, respectively. Al-Ouqaili and colleagues [8] reported that 92% and 96% of *K. pneumoniae* isolates were resistant to ceftazidime and cefotaxime, respectively, while the results of this study showed that the proportion of *K. pneumoniae* isolates resistant to the third-generation cephalosporins is higher than what was reported by Ullah et al. [21] in northwest Pakistan, where 54.35% of the isolates were resistant to ceftriaxone and ceftazidime.

Fluoroquinolone resistance has been developed in Enterobacterial isolates as a result of the widespread use of these drugs in recent years. Ahmadi et al. [22] concluded that *K. pneumoniae* is a leader among Enterobacteriaceae when it comes to developing resistance to several kinds of antibiotics, including quinolones. Clinicians treating infectious diseases throughout the world face a serious challenge from this multidrug-resistant opportunistic bacterium and showed that the highest resistance rates were observed against amoxicillin (98.2%), ceftriaxone (72.7%) and cefepime (72.7%), and that the rates of MDR and XDR isolates were 56.4% (*n* = 31) and 7.3% (*n* = 4), respectively, which agreed with our study.

Our result for ciprofloxacin resistance in *K. pneumoniae* isolates was (40.0%), and for levofloxacin was (50.0%), which were not in agreement with Jomehzadeh et al. [23] who showed different resistance ratios for ciprofloxcin (18.5%) and for levofloxacin (30.4%).

Despite the fact that antibiotic treatment has revolutionized the management of many infectious diseases, their increasing usage—in ways including inaccurate and indiscriminate prescribing, incorrect dosage and duration of therapy—and the over-the-counter distribution of antibiotics to the general public has supported the increase in resistance to antimicrobial agents [24]. Years ago, it was first proposed that a synthetic CRISPR/Cas system may be used as an antibiotic to eradicate particular bacterial genotypes [25]. Recent investigations have proven the capability of CRISPR/Cas to accurately eliminate drug resistance-related genes from bacterial strains in populations and to re-sensitize bacteria to antibiotics by deleting antimicrobial resistance (AMR)-encoding plasmids. Before CRISPR-Cas can be utilized to target AMR in wild microbial communities, there are still several obstacles to be addressed. To fully use the promise of this technology for reducing the environmental and clinical spread of AMR by mobile genetic elements (MGEs), it will be essential to identify a viable delivery strategy. The efficiency with which this may be accomplished will be substantially improved by straightforward CRISPR/Cas construct reprogramming to target specific genes of interest. Such a development could assist in combating reservoirs of resistance and even maintain or restore the antibacterial action of antibiotics. Individual bacterial strains were selectively eliminated from a population of mixed *Escherichia coli* genotypes by converting the population with a plasmid containing CRISPR/Cas designed to target a sequence particular to each genotype demonstrating the specificity of CRISPR/Cas antimicrobials. According to two studies, CRISPR-Cas9 may be given via phagemids (plasmids packed in phage capsids) to kill *E. coli* and *Staphylococcus aureus*, two clinically important bacterial pathogens, in a targeted manner. One of these investigations delivered CRISPR/Cas9 constructs tailored to target AMR genes harbored on plasmids by phagemid transduction, successfully removing these plasmids from bacteria. Additionally, conjugative plasmids were employed to transfer CRISPR/Cas9 to bacteria containing AMR genes in the chromosomal [26]. The other work revealed sequence-specific CRISPR/Cas9 delivery to bacteria harboring virulence genes and also proved that this method was capable of removing plasmids carrying AMR genes and successfully re-sensitizing bacteria to antibiotics [27].

In our study, we found diverse profiles for CRISPR/Cas systems in our isolate. The number, sequence and size of the CRISPR arrays are highly variable due to their variable exposure to different bacteriophages during their lifespan. This was in agreement with Makarova et al. [28], who revealed that the number, sequence and length of the CRISPR arrays are highly variable among bacterial species. Kuno and colleagues [29] demonstrated that high cyanophage diversity is also compatible with a huge variety of antiviral defense systems on the genome of *Microcystis aeruginosa.*

The CRISPR/Cas systems are involved in limiting the entrance of foreign DNA into bacteria and have also been implicated in their expression of virulence factors. These systems have been widely studied in several organisms, including pathogens and non-pathogens, but very few studies have actually demonstrated their in vivo activity [30]. Hence, and according to Bondy-Denomy and Davidson [30], information on the content of spacers, the proportion of them presenting concordance with a known sequence or of those that are unique to a specific species, is hard to find in publications, and would provide a fundamental point of view on the functionality of this system. These data will be useful in the interpretation of the functions of these systems in the strains being studied and should improve our understanding of bacterial evolution, as well as the impact of the horizontal transfer of genes in the environment and on human health. To the best of our knowledge, these systems have not yet been studied or characterized comprehensively in *K. pneumoniae*, which is among the top five pathogens causing nosocomial infections worldwide and belongs to the ESKAPE group [31,32,33,34,35,36]. *K. pneumoniae* easily disperses in hospital wards, contains diverse virulence factors and has large plasmids, conferring ecological advantages for its adaptation to several niches [37,38]. Therefore, we are interested to know whether *K. pneumoniae* has the CRISPR/Cas systems and whether these are related to horizontal gene transfer, multidrug resistance or virulence, or indeed otherwise.

The CRISPR/Cas system was found to provide protective immunity against viruses in prokaryotes, where spacers are acquired from invading elements and can mediate the immune reaction in a sequence-specific manner [39]. Therefore, the spacer repertoire could be a reflection of bacterial lifestyle [40].

To our knowledge, there are very limited studies in the Arab world, and this is the first report in our country, Iraq, which focused on the CRISPR-Cas system and its association with MDR, XDR and PDR *K. pneumoniae*, for which no or limited drugs can be used to treat infections caused by highly resistant strains. In this study, 32.0% (32/100) of *K. pneumoniae* isolates had CRISPR/Cas, which is considered a low proportion, whilst 13.0%, 24.0% and 17.0% had CRISPR1/Cas, CRISPR2/Cas and CRISPR3/Cas, respectively. The low prevalence of the CRISPR/Cas systems in this study could be attributed to the fact that the majority of the strains tested were resistant to multiple antibiotics; thus, they were found to be negative for these systems. When we compared our findings to other studies, Li and colleagues [41] noted that, overall, 30.7% of clinical *K. pneumoniae* isolates had the CRISPR/Cas system in Taiwan. In the work carried out by Wang and colleagues [42], CRISPR/Cas was detected in 21.32% (29/136) of *K. pneumoniae* isolates, 14 of which were positive for both CRISPR2 and CRISPR3, whilst other strains were positive for either CRISPR2 (*n* = 13) or CRISPR3 (*n* = 2). Zhou and colleagues [43] demonstrated that about one third of all 300 isolates of *K. pneumoniae* contained this system. By comparison, Lin et al.’s [44] recent study found that only 6 out of 52 isolates of *K. pneumoniae* had this system and, accordingly, it appears that the CRISPR/Cas system is not widely distributed in *K. pneumoniae.*

One of the strengths of the current study was that the results demonstrated the present findings of the CRISPR/Cas system and its association with the absence of antibiotic resistance genes. The CRISPR/Cas system was fully functional in *K. pneumoniae* isolates with accessible complete or draft genomes, indicating that CRISPR/Cas is not widespread in *K. pneumoniae* [45]. CRISPR/Cas was detected in 33/100 (33.0%) clinical strains of *K. pneumoniae* isolated from hospital settings. The incidence of the CRISPR/Cas system is variable, ranging from 30.7% (54/176) to 12.4% (27/217) [41,44].

An inverse correlation between the presence of the CRISPR/Cas system and antibiotic resistance to the different antimicrobial agents used was observed. Our study showed that all the isolates that contained this system had lower resistance to β-lactams, quinolones, aminoglycosides and β-lactams/enzyme inhibitors. Mackow and colleagues [46] revealed that the majority of *K. pneumoniae* isolates containing CRISPR/Cas were pan-sensitive, whereas those resistant to multiple antibiotics were observed in the isolates lacking CRISPR. Wang and colleagues [42] inferred that the presence of this system can be associated with lower drug resistance and to some extent, may prevent the acquisition of drug resistance genes in *K. pneumoniae*. Lin et al. [44] concluded that an inverse correlation between the presence of CRISPR/Cas loci and carbapenem resistance in *K. pneumoniae* was highly prevalent. Gholizadeh and colleagues [5] showed that *K. pneumoniae* isolates having the CRISPR/Cas system had a lower resistance to β-lactams, tetracyclines, quinolones, aminoglycosides and β-lactam inhibitors, which suggests an inverse correlation between the presence of CRISPR/Cas and antibiotic resistance.

The presence of either CRISPR or Cas in resistant bacteria can indicate that these systems were present before but were lost during evolution. They were lost in order to give the bacteria the chance to host AMR genes and become resistant [47]. The number of susceptible bacteria was small in this study, as this is a cross-sectional study, and the majority of the bacteria causing infections are MDR, XDR or PDR, which is very alarming. Nevertheless, we can conclude that the absence of CRISPR/Cas in these resistant bacteria supports our hypothesis about the inverse relation between antibiotic resistance and the presence of these systems. The diversity noted in the strains is also an interesting finding. This indicates that the continuous evolution of the bacteria and genetic recombination events is a part of the bacterial response to continuous exposure to antibiotics. This, indeed, suggests the need for more comprehensive genomic studies to examine the local as well as international strains.

## 4. Materials and Methods

### 4.1. Ethics Statement

This study was approved by the Medical Ethics Committee of the University of Al-Anbar Governorate, Ramadi, Iraq (approval number 9, 12 February 2022). Written informed consent was obtained from all patients or their guardians before participation in the study.

### 4.2. Study Design and Patients

This is a cross-sectional study conducted during the period between October 2021 and March 2022. Various clinical specimens were collected from inpatients admitted to the Medical City, Baghdad and Al Ramadi Teaching Hospitals, Ramadi, Iraq. A total of 100 strains of *K. pneumoniae* were isolated from various samples including 50% wound swabs (diabetic foot infections, osteomyelitis and burn infections), 30% urinary samples from catheterized units and 20% ear swabs from otitis media patients.

### 4.3. Isolation and Identification of K. pneumoniae Isolates

The clinical specimens were processed and cultured in the microbiology laboratories of the same hospitals which performed all bacteriological examinations and confirmatory biochemical tests [48]. All specimens were cultured on blood agar, MacConkey agar and Eosin Methylene blue agar (Merck, Darmstadt, Germany) and incubated at 37 °C for 24–48 h. *K. pneumoniae* isolates were identified using morphological and cultural criteria and standard biochemical tests including Gram stain, lysine iron agar (LIA), triple sugar iron (TSI) agar, ornithine decarboxylase and urea/citrate utilization [49]. The VITEK^®^2 Compact B System (BioMérieux, Marcy-l’Etoile, France) was used for the confirmative and final identification of *K. pneumoniae* isolates using VITEK^®^2 GN ID cards [6].

### 4.4. Antibiotic Susceptibility Testing (AST)

The AST of the collected isolates was determined via the VITEK^®^2 Compact B System (BioMérieux, Marcy-l’Étoile, France) using VITEK^®^2 AST-GN cards according to the manufacturer’s instructions. The results of the AST for the following antibiotics were recorded: piperacillin/tazobactam, cefazolin, cefoxitin, ceftazidime, ceftriaxone, cefepime, ertapenem, imipenem, amikacin, gentamicin, ciprofloxacin, levofloxacin, tigecycline and nitrofurantoin. The AST results from the VITEK^®^2 Compact B System (BioMérieux, Marcy-l’Étoile, France) were expressed as minimum inhibitory concentration (MIC) values and interpreted as susceptible, intermediate or resistant by reference to the Clinical and Laboratory Standard Institute (CLSI) guidelines [9,50]. The result of AST for tigecycline was interpreted according to the European Committee on Antimicrobial Susceptibility Testing (EUCAST) 2021 [9,51]. *Escherichia coli* (ATCC 25922) was used as a quality control strain for antibiotic susceptibility testing.

Based on antibiotic susceptibility profiles, the strains were classified as multidrug-resistant (MDR) if they were resistant to at least one antibiotic in three or more antimicrobial categories, extensively drug-resistant (XDR) if they were resistant to one antibiotic in ≥6 antimicrobial categories and pan-drug-resistant (PDR) if they were resistant to all antibiotics tested [52].

### 4.5. Phenotypic Tests for Detecting ESBL Production

NO45 cards (ESBL test panel) for the VITEK^®^2 Compact B System (BioMérieux, Marcy-l’Étoile, France) were used to test each isolate for ESBL production [53]. Six wells made up the panel, the first with cefepime (1.0 μg/mL) and the second, the third and the fourth with cefotaxime (0.5 μg/mL), ceftazidime (0.5 μg/mL) and cefepime/clavulanate (1/10 μg/mL), respectively. The fifth and the sixth wells contained cefotaxime/clavulanate (0.5/4 μg/mL) and ceftazidime/clavulanate (0.5/4 μg/mL), respectively. A spectrophotometric scanner was used to record bacterial growth. It was considered indicative of ESBL production when growth was reduced in cephalosporin plus clavulanate wells as compared with cephalosporin alone [53].

### 4.6. Phenotypic Tests for Detecting Metallo-Beta-Lactamases (MBL)

All the strains resistant to one or more carbapenems (*n* = 15) were tested. A concentration of 750 mg of EDTA-imipenem or meropenem was created by combining 10 μL of MBL inhibitor solution (0.5 M EDTA) with 10 μg of imipenem or meropenem disks. The disks were promptly dried before being placed in an airtight container without desiccant and kept at 4 °C. After being modified to meet the McFarland 0.5 turbidity criterion, bacterial isolates were spread onto Muller Hinton agar. Imipenem or meropenem (10 μg) disks or imipenem or meropenem with 750 μg EDTA solution were placed on Muller Hinton agar. As a control, a different disk with only 750μg of EDTA was likewise positioned. After overnight incubation, the isolate was deemed to be an MBL-producer if there was a smaller than 7 mm inhibitory zone difference between carbapenem disk and carbapenem-EDTA disk [54,55].

### 4.7. Detection of CRISPR/Cas Genes

Genomic DNA was extracted using the SaMag TM Bacterial DNA extraction kit using SaMag-12^TM^ automatic nucleic acid extraction system (SaMag, Cepheid, Buccinasco, Italy) according to the manufacturer’s instructions. The Quantus^TM^ Fluorometer (Promega, Madison, WI, USA) was used to measure the concentration of extracted nucleic acid to detect the quality and quantity of the samples for further applications [56].

CRISPR/Cas genes were detected via PCR using the previously described specific primers [9] shown in Table 3.

The PCR cycle for Cas1 and Cas3 was as follows: initial denaturation at 95 °C for 5 min, denaturation at 94 °C for 1 min, annealing at 60 °C for 30 s and extension at 72 °C for 1 min. The denaturation, annealing and extension steps were repeated for 35 cycles with a final extension step at 72 °C for 10 min. The PCR cycle for all CRISPR types were as follows: initial denaturation at 95 °C for 5 min, denaturation at 94 °C for 1 min, annealing at 62 °C for 30 s and extension at 72 °C for 1 min. The denaturation, annealing and extension steps were repeated for 35 cycles with a final extension step at 72 °C for 10 min. A negative control was included in all the experiments to ensure sterility. All PCR products were then separated and detected via gel electrophoresis in a 1.5% agarose gel at 50 V for 5 min and then at 100 V for 1 h in a 1X Tris/Borate/EDTA (TBE) buffer containing the DNA safe stain. Finally, the PCR product size was correlated with a 100 base-pair DNA ladder (Fermentas, Maryland, USA) to confirm the expected PCR amplicon band via Ultraviolet Transilluminator (Vilber lourmat, Marne-la-Vallée cedex 3, France.).

### 4.8. Statistical Analysis

The Statistical Package for the Social Sciences (SPSS) version 20.0 (IBM Corporation, Armonk, NY, USA) was used to analyze the data in terms of mean, percentage and frequency. Chi-squared and Fisher’s exact tests were used to evaluate the significant associations between variables (*p*-value 0.05).

## 5. Conclusions

Due to the low prevalence of the CRISPR/Cas systems in antibiotic resistant *K. pneumoniae* isolates, we conclude that these systems can offer protection against exogenous antibiotic resistance which can be acquired in the absence of CRISPR/Cas modules. This is, indeed, important as these systems can be developed in the future to be used as tools to fight antibiotic-resistant bacteria. The CRISPR/Cas system provides new opportunities to eradicate MDR strains, as this RNA-guided DNA nuclease can specifically cleave bacterial genes, leading to the re-sensitization of the antibiotic-resistant cells. More studies are needed in the future using bacterial whole genome sequencing to delineate the relation of these systems with various genomic contents of the bacteria, and for a better understanding of the contribution of these systems to antibiotic resistance in multiple species of bacteria. Furthermore, an investigation of the associations between the CRISPR system’s existence with the epidemiological and pathogenicity factors of miscellaneous bacterial pathogens is needed in future studies.

## Figures and Tables

**Figure 1 antibiotics-12-00980-f001:**
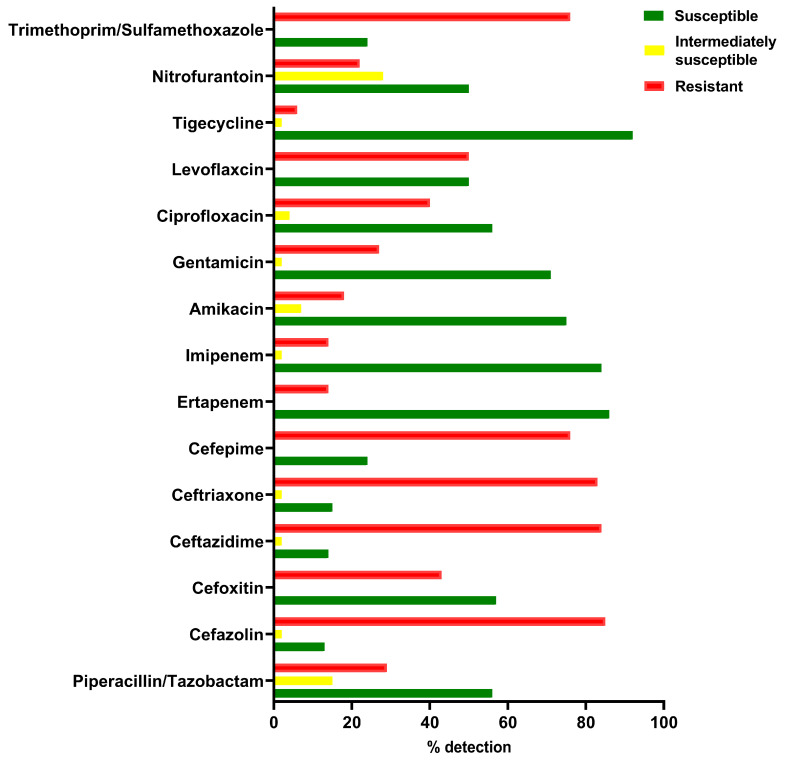
Summary of antimicrobial susceptibility profile (%) for *K. pneumoniae* isolates depending on MIC values.

**Figure 2 antibiotics-12-00980-f002:**
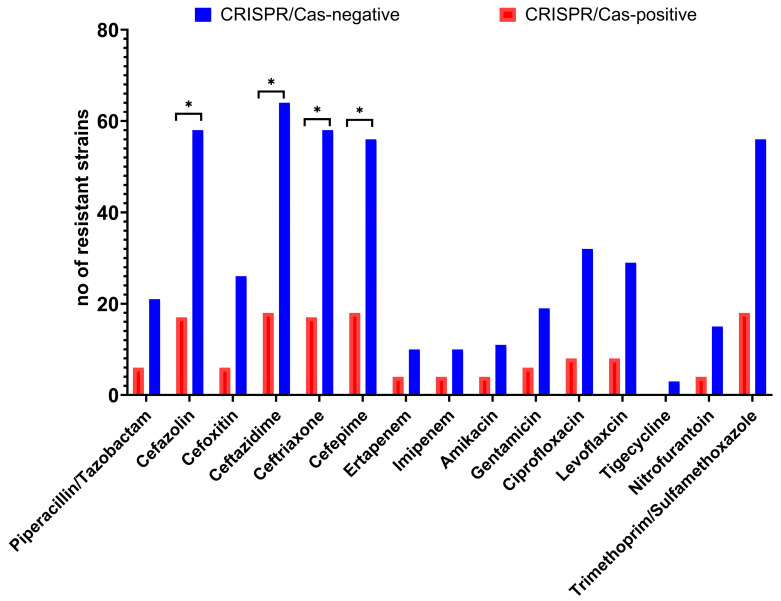
Drug resistance patterns between CRISPR/Cas-positive and -negative isolates of *K. pneumoniae*. *: significant difference (*p* < 0.05).

**Table 1 antibiotics-12-00980-t001:** Summary of the results of PCR for detecting Cas and CRISPR genes. The number of strains having one or both genes, and those devoid of any gene, are shown.

Profile	CRISPR1	CRISPR2	CRISPR3	Cas 1	Cas 3	No. of Strains
**1**						**19**
**2**						**10**
**3**						**7**
**4**						**4**
**5**						**2**
**6**						**6**
**7**						**16**
**8**						**4**
**9**						**3**
**10**						**1**
**11**						**1**
**12**						**1**
**13**						**1**
**14**						**1**
**15**						**1**
**16**						**2**
**17**						**2**
**18**						**1**
**19**						**1**
**20**						**2**
**21**						**3**
**22**						**9**
**23**						**3**


 Negative; 

 Positive.

**Table 2 antibiotics-12-00980-t002:** The association between MDR, XDR, PDR and prevalence of CRISPR/Cas systems.

MDR/XDR/PDR	Number of Strains	CRISPR/Cas-Positive No. (%)	CRISPR/Cas-Negative No. (%)
**Non-MDR**	14 (14%)	8 (57.14)	6 (42.86)
**MDR ^1^**	71 (71%)	20 (28.17)	51 (71.83)
**XDR ^2^**	11 (11%)	2 (18.18)	9 (81.82)
**PDR ^2^**	4 (4%)	2 (50)	2 (50)
**CR ^3^**	15	4	11

Non-MDR refers to single drug-resistant or non-resistant to any drug. MDR refers to multidrug-resistant, XDR refers to extensively drug-resistant, PDR refers to pan drug-resistant. ^1^ all the MDR strains were ESBL producers; ^2^ XDR and PDR included only metallo-beta-lactamase producers (CR); ^3^ combinations of XDR and PDR.

**Table 3 antibiotics-12-00980-t003:** The sequences of primers used in this study for the detection of CRISPR/Cas genes.

Gene	Primer Sequence (5′ → 3′)	PCR Product Size (bp)
Cas 1	F-GCTGTTTGTCAAAGTTACCCGCGAACTC	208
	R-GTTTTGATCGCCTCATGAGTCACAGTTG	
Cas 3	F-TGGCCGACATTTGATTCAGC	620
	R-CCATGCTTAACATTCATCAC	
CRISPR 1	F-CAGTTCCTGCAACCTGGCCT	Variable
	R-CTGGCAGCAGGTGATACAGC	
CRISPR 2	F-GTAGCGAAACCCTGATCAAGCG	Variable
	R-GCGCTACGTTCTGGGGATG	
CRISPR 3	F-GACGCTGGTGCGATTCTTGAG	Variable
	R-CGCAGTATTCCTCAACCGCCT	

## Data Availability

The data presented in this study are available on request from the corresponding author.

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
