# Peer review of "Inverse Association between the Existence of CRISPR/Cas Systems with Antibiotic Resistance, Extended Spectrum β-Lactamase and Carbapenemase Production in Multidrug, Extensive Drug and Pandrug-Resistant Klebsiella pneumoniae"

_antibiotics, 2023, doi:10.3390/antibiotics12060980_

Round 1
Reviewer 1 Report
Dear Authors
Thanks for your manuscript with title: nverse association between the existence of CRISPR/Cas systems with antibiotic resistance, extended spectrum β-lactamase and carbapenemase production in multidrug, extensive drug and pandrug resistant Klebsiella pneumoniae. The review of the aforementioned manuscript has been finished and there are some points about it wich you could find at the attached file.
Best Regards

I think the quality of english language is good and need some minor editing.
Author Response
Reviewer 1
Thanks for your manuscript with title: nverse association between the existence of CRISPR/Cas systems with antibiotic resistance, extended spectrum β-lactamase and carbapenemase production in multidrug, extensive drug and pandrug resistant Klebsiella pneumoniae. The review of the aforementioned manuscript has been finished and there are some points about it wich you could find at the attached file.
Best Regards
in the attached file :
- This results are repeat (Figure 1) and don't needed
Response : the section was revised as follows (highlighted in yellow - line number ??):
As shown in Figure 1, majority of the isolates were resistant to multiple beta-lactam drugs including carbapenems and beta-lactam/beta lactamase inhibitors. Furthermore, resistance was common to other drug such as fluoroquinolones and aminoglycosides. The majority of the strains were susceptible to tigecycline, with only a few resistant strains (6%).
- Please write the name of bacteria in italic format
Response: done
- Please bring the name of bacteria in full for the fist time and use abbreviate for the next time. pneumoniae
Response: done
- Is it normal that more than of the isolates obtained from wound samples? As you know Klebsiella pneumoniae is not a common agent in comparison to Staphylococcus aureus and Pseudomonas aeruginosa
Response: the samples were obtained from the microbiology labs in this cross-sectional study. The focus was more on resistant bacteria, especially MDR, XDR and PDR. Probably other sources of infection did not yield resistant bacteria thus were not shared by these labs.
- What about the quality control of the AST?
Response: this was added to the methods:
“Escherichia coli (ATCC 25922) was used as a quality control strain for antibiotic susceptibility testing.”
- how could you checked and evaluated the results?
Response: A standard method was used and MBL production was confirmed by PCR (data not shown).
- What was your quality control on PCR?
Response: We add the following sentence to the manuscript:
Negative control was included in all the experiments to ensure sterility.
- The figure in s1 need to explain about the wells and what were the Positive and negative control? How did you confirm the PCR product?
Response: We used negative control was included in all the experiments to ensure sterility. We depended on the PCR product size to confirm the PCR results, and the samples with positive results were repeated to make sure that we have true positive results.
Thank you very much for your comments
Reviewer 2 Report
The work addresses an important issue highlighting the inverse relationship between the presence of CRISPR/Cas systems in K. pneumoniae. The authors were able to demonstrate that multidrug resistance in ESBL-producing and carbapenem-resistant K. pneumoniae occurred more frequently in strains negative for CRISPR/Cas system. The idea of the research is one of the attractive topics discussed nowadays and it paves the way towards new strategies in combating the multidrug resistance issue. However, there are a few issues that the authors should address prior the publication of the manuscript, as follows:
1. Throughout the manuscript: The authors are writing “Klebsiella pneumoniae” in full name each time it is mentioned. It should be written in full only the first mentioning then it should be “K. pneumoniae” afterwards everywhere. Please check carefully and correct.
2. Introduction: Line: 77: The percentage of carbapenem resistance in line 77 (14%) is different from that in the abstract Line 20 and Lines 85-86 in Introduction (15%) and in the supplementary table S2 the percentage of carbapenem resistance could be calculated as 15% and not 14%. This is important when calculating the percentage of CRISPR/Cas systems among carbapenem-resistant strains in Lines 120-121. Please correct.
3. Introduction: Line 75: “inducing” should be corrected to “including”.
4. Introduction: Line: 85: This is the first mention of “MDR” the authors should write it in full. Accordingly, this abbreviation which stands for “multidrug-resistant” can not be used for “multidrug resistance” mentioned in line 260 in discussion. Please correct.
5. Results: Lines 87-88: “From Only 14% (14/100) of the strains were 87 susceptible to different antimicrobial agents” The sentence is not understood. What do the authors mean?
6. Results: Lines 121-122: “and had higher resistance to other antibiotics,”. Please clarify.
7. Results: Line 123: “drug-sensitive strains” For a strain to be a sensitive one, it has to show sensitivity against the tested antibiotics. Looking at the supplementary Table S2, it is obvious that some of the strains referred to by the authors as “sensitive” are not (Isolates 1, 2, 4, 6,10, and 12).. these isolates show intermediate resistance or even resistance to one or more of the tested antibiotics. Accordingly, these are not sensitive strains. The authors could name the strains as “non-MDR” as they mentioned in Table S2 or the authors could classify the strains differently: as sensitive (with no resistance at all) and non-MDR; but to name resistant strains as “sensitive” while they are not, is not possible. Please correct.
8. Figure 2 legend: Line 147: “It is obvious that absence of CRISPR is more common among resistant strains, even if non-significant”. This is a conclusion sentence that should be within the results or discussion but not in a figure’s legend.
9. Discussion: Line 177: “E. coli” This is the first mention of this organism. Please write the name of the organism in full.
10. Discussion: Lines 194-200: “The high level of resistant isolates corresponds to many local and international studies”. Since the authors are going to compare their study to those published internationally or locally, they have to mention the country and the year when these studies have been conducted. In epidemiology, we have to state where and when… Otherwise mentioning “results were similar to those observed by Khalaf and Al-Ouqaili [8] who concluded that the rates of resistance to ceftriaxone and ceftazidime were 88% and 84% respectively” does not indicate whether it is locally? Internationally? Recent study? Old one? Please adjust the paragraph.
11. Throughout the manuscript: Before the word” respectively” a comma should exist.
12. Throughout the manuscript: drug resistant should be written “drug-resistant” eg: Line 33-34, 354, 355…..
13. Discussion: Line 202: “Third generation” of which drug? Cephalosporins and fluoroquinolones are classified by generation… Please specify.
14. Discussion: Line 203: “the proportion of resistant Klebsiella isolates of this generation” “to this generation”.
15. Discussion: Lines 214, 217: The abbreviations “AMR” and “MGE” are mentioned here for the first time. Please write the complete words for these abbreviations.
16. Discussion: Lines 214-216: “Before CRISPR-Cas can be utilized to target AMR in wild microbial communities. There are still several obstacles to be addressed.” This is one sentence and not two. Please correct.
17. Discussion: Line 235: “cas” should be capitalized.
18. Discussion: Line 262: “n” is not understood.
19. Discussion: Line 266: “concentrate” may be better to use “focus on” or any other verb as “concentrate” implies a chemical concentration.
20. Discussion: Line 272: “we” is meaningless in the sentence. Please rephrase.
21. Discussio: Line 277-278: “whilst 13 isolates were CRISPR2-positive, and two isolates were CRISPR3-positive” Do the authors mean 13 and 2 isolates in their study? Please clarify.
22. Discussion: Lines: 285-287:” The CRISPR/Cas system was fully functional in Klebsiella pneumoniae isolates with accessible complete or draft genomes, indicating that CRISPR/Cas is not widespread in Klebsiella pneumoniae” Please explain this sentence.
23. Conclusion: Line 414: “The CRISPR/Cas9” Why do the authors emphasize the role of Cas9 while not being mentioned before? The whole study is talking about Cas1 and Cas3.
The authors need to adjust several comments that have been mentioned in my review enabling them to avoid mistakes in the English language.
Author Response
Comments and Suggestions for Authors
The work addresses an important issue highlighting the inverse relationship between the presence of CRISPR/Cas systems in K. pneumoniae. The authors were able to demonstrate that multidrug resistance in ESBL-producing and carbapenem-resistant K. pneumoniae occurred more frequently in strains negative for CRISPR/Cas system. The idea of the research is one of the attractive topics discussed nowadays and it paves the way towards new strategies in combating the multidrug resistance issue. However, there are a few issues that the authors should address prior the publication of the manuscript, as follows:
- Throughout the manuscript: The authors are writing “Klebsiella pneumoniae” in full name each time it is mentioned. It should be written in full only the first mentioning then it should be “ pneumoniae” afterwards everywhere. Please check carefully and correct.
Response: done as advised by the reviewer
- Introduction: Line: 77: The percentage of carbapenem resistance in line 77 (14%) is different from that in the abstract Line 20 and Lines 85-86 in Introduction (15%) and in the supplementary table S2 the percentage of carbapenem resistance could be calculated as 15% and not 14%. This is important when calculating the percentage of CRISPR/Cas systems among carbapenem-resistant strains in Lines 120-121. Please correct.
Response: the % was unified in all the manuscript as 15% , sorry for the mistake.
- Introduction: Line 75: “inducing” should be corrected to “including”.
Response: revised and corrected
- Introduction: Line: 85: This is the first mention of “MDR” the authors should write it in full. Accordingly, this abbreviation which stands for “multidrug-resistant” can not be used for “multidrug resistance” mentioned in line 260 in discussion. Please correct.
Response: revised and corrected, MDR was defined as “multidrug-resistant”, but for the discussion we removed MDR, as we meant here multidrug resistance, and not resistant
- Results: Lines 87-88: “From Only 14% (14/100) of the strains were 87 susceptible to different antimicrobial agents” The sentence is not understood. What do the authors mean?
Response: sorry for the mistake, “From” was removed from the sentence.
- Results: Lines 121-122: “and had higher resistance to other antibiotics,”. Please clarify.
Response: the sentence was revised as follows:
The prevalence of CRISPR/Cas system was 4 (26.67%) in carbapenem-resistant strains and had higher resistance to other antibiotics as these strains were either XDR, or PDR.
- Results: Line 123: “drug-sensitive strains” For a strain to be a sensitive one, it has to show sensitivity against the tested antibiotics. Looking at the supplementary Table S2, it is obvious that some of the strains referred to by the authors as “sensitive” are not (Isolates 1, 2, 4, 6,10, and 12).. these isolates show intermediate resistance or even resistance to one or more of the tested antibiotics. Accordingly, these are not sensitive strains. The authors could name the strains as “non-MDR” as they mentioned in Table S2 or the authors could classify the strains differently: as sensitive (with no resistance at all) and non-MDR; but to name resistant strains as “sensitive” while they are not, is not possible. Please correct.
Response: revised and corrected , these strains were defined to be without MDR phenotype, i.e susceptible to most antibiotics, and possibly resistant to less than 3 drugs.
- Figure 2 legend: Line 147: “It is obvious that absence of CRISPR is more common among resistant strains, even if non-significant”. This is a conclusion sentence that should be within the results or discussion but not in a figure’s legend.
Response: The sentence was removed form figure legend and moved to results
- Discussion: Line 177: “E. coli” This is the first mention of this organism. Please write the name of the organism in full.
Response: revised and corrected, the name was written in full as Escherichia coli
- Throughout the manuscript: Before the word” respectively” a comma should exist.
Response: revised and corrected
- Throughout the manuscript: drug resistant should be written “drug-resistant” eg: Line 33-34, 354, 355…..
Response: revised and corrected
- Discussion: Line 202: “Third generation” of which drug? Cephalosporins and fluoroquinolones are classified by generation… Please specify.
Response: revised and corrected, “cephalosporins” was added after third generation
- Discussion: Line 203: “the proportion of resistant Klebsiella isolates of this generation” “to this generation”.
Response: the sentence was revised as follows:
While the results of this study showed that the proportion of resistant K. pneumoniae isolates to the third generation cephalosporins is higher than what was reported by Ullah et al., [21] in northwest Pakistan, where 54.35% of the isolates were resistant to ceftriaxone and ceftazidime.
- Discussion: Lines 214, 217: The abbreviations “AMR” and “MGE” are mentioned here for the first time. Please write the complete words for these abbreviations.
Response: The abbreviations were defined as follows: AMR for antimicrobial resistance , and MGEs for mobile genetic elements
- Discussion: Lines 214-216: “Before CRISPR-Cas can be utilized to target AMR in wild microbial communities. There are still several obstacles to be addressed.” This is one sentence and not two. Please correct.
Response: the sentence was revised as follows: “Before CRISPR-Cas can be utilized to target AMR in wild microbial communities, there are still several obstacles to be addressed”.
- Discussion: Line 235: “cas” should be capitalized.
Response: done
- Discussion: Line 262: “n” is not understood.
Response: sorry for the typo error, here we meant “in”, it was corrected
- Discussion: Line 266: “concentrate” may be better to use “focus on” or any other verb as “concentrate” implies a chemical concentration.
Response: The word “concentrate” was replaced by focused
- Discussion: Line 272: “we” is meaningless in the sentence. Please rephrase.
Response: The word “we” was replaced by were
- Discussion: Line 277-278: “whilst 13 isolates were CRISPR2-positive, and two isolates were CRISPR3-positive” Do the authors mean 13 and 2 isolates in their study? Please clarify.
No, here we are referring to the work by Wang and colleagues [40] as CRISPR/Cas was detected in 21.32% (29/136) of the isolates of K. pneumoniae, 14 of which were CRISPR2- and CRISPR3-positive, whilst 13 isolates were CRISPR2-positive, and two isolates were CRISPR3-positive.
But we revised this part to make it clearer as follows:
“In the work done by Wang and colleagues [40] CRISPR/Cas was detected in 21.32% (29/136) of K. pneumoniae isolates, 14 of which were positive for both CRISPR2 and CRISPR3, whilst other strains were positive for either CRISPR2 (n=13) or CRISPR3 (n=2).”
Discussion: Lines: 285-287:” The CRISPR/Cas system was fully functional in Klebsiella pneumoniae isolates with accessible complete or draft genomes, indicating that CRISPR/Cas is not widespread in Klebsiella pneumoniae” Please explain this sentence.
Response: Revision done
- Conclusion: Line 414: “The CRISPR/Cas9” Why do the authors emphasize the role of Cas9 while not being mentioned before? The whole study is talking about Cas1 and Cas3.
Response: thanks for the comment, 9 was removed from the conclusion.
- The current study is interesting; however, the authors should address the following comments to improve the quality of the manuscript.
Thank you very much for your comments
Reviewer 3 Report
Inverse Association between the Existence of CRISPR/Cas Sys-2 tems with Antibiotic Resistance, Extended Spectrum 3 β-lactamase and Carbapenemase Production in Multidrug, Ex-4 tensive Drug and Pandrug Resistant Klebsiella pneumoniae
Manuscript ID: antibiotics-2392919
- The current study is interesting; however, the authors should address the following comments to improve the quality of the manuscript:
1. Please write the scientific names of bacterial pathogens and genes in the correct form all over the manuscript and in the References section (should be italic).
2. The abstract must illustrate the used methods and the most prevalent results (give more hints about methods and results).
3. Discuss more significant results of this study.
4. Please discuss the more alarming results of this research in the discussion section.
5. Recheck all numbers and presents in the text to ensure their accuracy.
6. Some references are not based on the journal guidelines. Please re-check all references.
7. Minor spacing problems exist between the words in some parts of the text.
8. The English of the manuscript should be reviewed and syntax and errors should be corrected before publication.
9. Use the following valuable studies performed on pathogen isolates in the introduction or discussion section and add related references, including:
https://doi.org/10.4103/bbrj.bbrj_302_22
https://doi.org/10.1002/jcla.24342
Inverse Association between the Existence of CRISPR/Cas Sys-2 tems with Antibiotic Resistance, Extended Spectrum 3 β-lactamase and Carbapenemase Production in Multidrug, Ex-4 tensive Drug and Pandrug Resistant Klebsiella pneumoniae
Manuscript ID: antibiotics-2392919
- The current study is interesting; however, the authors should address the following comments to improve the quality of the manuscript:
1. Please write the scientific names of bacterial pathogens and genes in the correct form all over the manuscript and in the References section (should be italic).
2. The abstract must illustrate the used methods and the most prevalent results (give more hints about methods and results).
3. Discuss more significant results of this study.
4. Please discuss the more alarming results of this research in the discussion section.
5. Recheck all numbers and presents in the text to ensure their accuracy.
6. Some references are not based on the journal guidelines. Please re-check all references.
7. Minor spacing problems exist between the words in some parts of the text.
8. The English of the manuscript should be reviewed and syntax and errors should be corrected before publication.
9. Use the following valuable studies performed on pathogen isolates in the introduction or discussion section and add related references, including:
https://doi.org/10.4103/bbrj.bbrj_302_22
https://doi.org/10.1002/jcla.24342
Author Response
- Please write the scientific names of bacterial pathogens and genes in the correct form all over the manuscript and in the References section (should be italic).
Response: done
- The abstract must illustrate the used methods and the most prevalent results (give more hints about methods and results).
Response: done
- Discuss more significant results of this study.
Response: done
- Please discuss the more alarming results of this research in the discussion section.
Response: done
- Recheck all numbers and presents in the text to ensure their accuracy.
Response: done
- Some references are not based on the journal guidelines. Please re-check all references.
Response: done
- Minor spacing problems exist between the words in some parts of the text.
Response: done
- The English of the manuscript should be reviewed and syntax and errors should be corrected before publication.
Response: done
- Use the following valuable studies performed on pathogen isolates in the introduction or discussion section and add related references, including:
Response: done, we are add the following references:-
https://doi.org/10.4103/bbrj.bbrj_302_22
https://doi.org/10.1002/jcla.24342
Thank you very much
Reviewer 4 Report
Dear Author/Editor;
The manuscript's topic is timely and will be of interest to the journal readers.
Also, the manuscript is very well written, and the ideas flow logically. The review of the literature is thorough, so the reader is given an adequate background about the topic. Also, " Inverse Association between the Existence of CRISPR/Cas Sys- 2 tems with Antibiotic Resistance, Extended Spectrum 3 β-lactamase and Carbapenemase Production in Multidrug, Ex- 4 tensive Drug and Pandrug Resistant Klebsiella pneumoniae" has been assessed by me. Although it is of interest, we are not able to consider it for publication in its current form. I have raised several points which we believe would improve the manuscript and may allow the current version to be published in the antibiotics.
Pls, see the attached files
1: One hundred isolates of Klebsiella pneumoniae were isolated from a variety of 70 clinical specimens.? Where is locations? İf possible pls provide GPS locations and ethical letter?
2: cefazolin (85%), ceftazidime (84%), ceftriaxone (83%), cefepime (76%), cefoxitin (43%), 76 including some strains resistant to both ertapenem & imipenem (14%), and be- 77 -lactam/beta lactamase inhibitors (piperacillin tazobactam ; 29%). …. Pls provide detail information cefazolin, ceftazidime etc? Which brand or code etc?
3: Pls add future studies?
5 Figure must be more clear n clean. Should indicate size of DNA fragmen
Author Response
Reviewer 4
The manuscript's topic is timely and will be of interest to the journal readers.
Also, the manuscript is very well written, and the ideas flow logically. The review of the literature is thorough, so the reader is given an adequate background about the topic. Also, " Inverse Association between the Existence of CRISPR/Cas Sys- 2 tems with Antibiotic Resistance, Extended Spectrum 3 β-lactamase and Carbapenemase Production in Multidrug, Ex- 4 tensive Drug and Pandrug Resistant Klebsiella pneumoniae" has been assessed by me. Although it is of interest, we are not able to consider it for publication in its current form. I have raised several points which we believe would improve the manuscript and may allow the current version to be published in the antibiotics.
Pls, see the attached files
1: One hundred isolates of Klebsiella pneumoniae were isolated from a variety of 70 clinical specimens.? Where is locations? İf possible pls provide GPS locations and ethical letter?
Response: no 100 Klebsiella pneumoniae were isolated from a variety of 100 clinical specimens. This is mentioned in the methods
Also study location is mentioned:
“Various clinical specimens were collected from inpatients admitted to the Medical City, Baghdad and Al Ramadi Teaching Hospitals, Ramadi, Iraq.”
2: cefazolin (85%), ceftazidime (84%), ceftriaxone (83%), cefepime (76%), cefoxitin (43%), 76 including some strains resistant to both ertapenem & imipenem (14%), and be- 77 -lactam/beta lactamase inhibitors (piperacillin tazobactam ; 29%). …. Pls provide detail information cefazolin, ceftazidime etc? Which brand or code etc?
Response: it is mentioned in the methods that we used , this is the brand: BioMérieux, Marcy-l’Etoile, France
3: Pls add future studies?
Response: added
4: Figure must be more clear n clean. Should indicate size of DNA fragment
Response: done
Round 2
Reviewer 1 Report
Dear Authors
Thanks for revised manuscript. I think you addressed almost the reviewer comments.
Best Regards